# Ginseng Nanosizing: The Second Spring of Ginseng Therapeutic Applications

**DOI:** 10.3390/antiox14080961

**Published:** 2025-08-05

**Authors:** Jian Wang, Huan Liu, Xinshuo Ding, Tianqi Liu, Qianyuan Li, Runyuan Li, Yuan Yuan, Xiaoyu Yan, Jing Su

**Affiliations:** Key Laboratory of Pathobiology, Department of Pathophysiology, Ministry of Education, College of Basical Medical Sciences, Jilin University, 126 Xinmin Street, Changchun 130021, China; wangj24@mails.jlu.edu.cn (J.W.); liuhuan24@mails.jlu.edu.cn (H.L.); dingxs9922@mails.jlu.edu.cn (X.D.); liutq9922@mails.jlu.edu.cn (T.L.); liqy24@mails.jlu.edu.cn (Q.L.); lry23@mails.jlu.edu.cn (R.L.); yuany23@mails.jlu.edu.cn (Y.Y.)

**Keywords:** ginseng-derived vesicle-like nanoparticles, gintonin, ginsenoside, nanoparticles, engineering transformation, nanosizing

## Abstract

Plant-derived vesicles offer several advantages, including high yield, low cost, ethical compatibility, safety, and potential health benefits. These advantages enable them to overcome technological limitations associated with vesicles of mammalian origin. Ginseng, a prominent example of a natural botanical plant, is known for its abundant bioactive components. Recent studies confirmed that ginseng-derived vesicles offer significant advantages in the treatment of human diseases. Therefore, this study reviews the extraction and purification processes of ginseng-derived vesicle-like nanoparticles (GDVLNs), their therapeutic potential, and the active ingredients in GDVLNs that may exert pharmacological activities. Furthermore, this study evaluates the research and applications of nanosized ginseng extracts, with a primary focus on ginsenosides.

## 1. Introduction

Since Halperin et al. first observed vesicular granules in carrot cells using electron microscopy [1] and Canal et al. isolated extracellular vesicles (EVs) from the extracellular fluid of sunflowers in 2009 [2], several studies have confirmed that plant cells, like animal cells, can secrete vesicles into the interstitial space of cells. Thus, similar to animal vesicles, EVs in plants play important roles in the transport of secondary metabolites and other biosignaling molecules, defense against pathogens, and messaging [3]. Studies have confirmed the importance of EVs in various diseases and their potential therapeutic applications. This raises several questions regarding the biological relevance of plant-derived extracellular vesicles (PDEVs) to human health. One of the many advantages PDEVs have over mammal-derived EVs (MDEVs) is that they can be produced in large quantities for future therapeutic applications in a cost-effective manner [4], without the need for culture media and serum to prepare a sufficient cell volume for the isolation of MDEVs. For example, the volume of conditioned medium required for the 4.03 × 10^12^ EVs obtained from mesenchymal stem cells (MSCs) by Börger et al. was approximately 4300 mL [5]. In contrast, approximately 1.3932 × 10^15^ EVs could be isolated from the same weight of carrots [6]. Considering that the large-scale production of EVs is crucial for practical clinical applications, PDEVs will have a significant advantage in future practical applications owing to their affordability and lower time cost. Another advantage of PDEVs is their safety. Unlike animals, plants do not carry zoonotic or human pathogens [7]. PDEVs are similar to MDEVs in properties such as size distribution, surface charge, morphology, and density [8]. Another unique advantage of PDEVs is that they often carry specific bioactive metabolites involved in primary and secondary metabolism [9]. Therefore, active ingredient-enriched PDEVs may have a more comprehensive therapeutic effect than a single extract. Consequently, studies on PDEVs are increasing, gradually demonstrating their excellent potential as drug carriers and biotherapeutics.

Ginseng (Panax ginseng C. A. Mey.) is a perennial herbaceous plant belonging to the genus *Panax* in the family *Pentaphyllaceae*. Ginseng has been demonstrated to possess anti-tumor, anti-inflammatory, antioxidant, and immune system-balancing properties [10]. The bioactive compounds present in ginseng, including ginsenosides, polysaccharides, and phenols, contribute significantly to its therapeutic properties [11]. In recent years, researchers have isolated vesicular particles from ginseng roots and demonstrated their excellent therapeutic efficacy in a variety of diseases in several studies. These vesicular particles are encapsulated in membranes with nanosized active ingredients that enhance their targeting and biocompatibility to specific organs, while mitigating the potential toxicity and side effects associated with high-dose use. Recent research suggests that vesicular particles from ginseng root have unexpected effects on the homeostasis of our entire body, and that “nanosizing” ginseng may be a promising new direction for future ginseng applications.

A literature search for this review was based on the Web of Science database and covered a period from the publication of the first relevant literature in the field to the time of writing. The search terms include “ginseng-derived nanoparticles,” “ginsenoside nanoparticles,” “ginseng-derived exosomes,” “gintonin,” etc., in order to comprehensively cover the extraction and purification of GDVLNs, their active ingredients, applications, and studies related to the nanosensitization of ginseng extracts. The inclusion criteria were original studies and reviews directly related to the above research topics. We did not actively set up exclusion or screening criteria, but only naturally filtered the irrelevant literature by search terms to ensure a comprehensive overview of the research progress in this field.

Notably, the nomenclature of ginseng-derived vesicles has been inconsistent across studies. According to the nomenclature recommendations for EVs in the Basic Information on Extracellular Vesicle Research, unless vesicles isolated from plants meet the minimum requirements for the definition of EVs, a more general terminology is recommended [12]. Currently, the nomenclature of plant vesicles mainly depends on the use of tissue disruption techniques during isolation. Typically, vesicles isolated from plant exosomes by tissue infiltration are referred to as EVs, whereas vesicles isolated from disrupted plant tissues are referred to as exosome- or vesicle-like nanoparticles (NPs) [13]. Because ginseng root is often used as the main source of ginseng vesicles, high-speed centrifugation after tissue fragmentation is commonly used to isolate vesicles in experiments; therefore, GDVLNs will be used to refer to ginseng vesicles in this review.

## 2. Acquisition of GDVLNs

Common methods used in the isolation of PDEVs include ultracentrifugation after fragmentation, polyethylene glycol (PEG) precipitation, ultrafiltration, and size-exclusion chromatography. The dense tissue structure of ginseng roots facilitates the extraction of GDVLNs through tissue disruption followed by ultracentrifugation. Ginseng roots are typically crushed in a blender to obtain sap. Cell debris or organelles are then removed by two or three steps of low-speed centrifugation, followed by one or two ultra-high-speed centrifugation steps to enrich GDVLNs. Through a series of centrifugation cycles with varying centrifugal forces and durations, vesicles can be sequentially separated from the suspension based on the physical properties (e.g., density and size) of the GDVLNs [14] (Figure 1). However, several studies have shown structural differences in extracted GDVLNs. For example, Cao et al. obtained GDVLNs with an average diameter of 344.8 nm using sucrose density gradient centrifugation [15], while Cho et al. obtained GDVLNs with an average diameter of 92.04 ± 4.85 nm using the same method [16]. Additionally, using transmission electron microscopy, Xu et al. observed that GDVLNs have a typical cup-shaped structure [17], whereas Han et al. found that GDVLNs are spherical [18]. This discrepancy may be due to differences in specific parameters such as centrifugation force, time, centrifugation buffer, and temperature. Li et al. found that placing an appropriate amount of Optiprep buffer pad, a high-density isotonic material, at the bottom effectively improved the yield and quality of nanovesicles extracted from ginger. The particles purified using the Optiprep buffer pad showed a cleaner background, less aggregation, and better retention of their spherical morphology [19]. This suggests that the addition of a buffer layer during the extraction of GDVLNs can eliminate structural disruption and aggregation during ultra-high-speed centrifugation. Polymer precipitation has been suggested as a potentially effective separation method, particularly for commercial extraction. PEG-based nanovesicle isolation methods maintain the integrity of nanovesicles by entrapping them in a mesh structure, thereby avoiding disruption of the morphology of nanovesicles by prolonged high-speed centrifugation [20]. Exploring the effect of varying extraction parameters on the functional activity of GDVLNs is a promising avenue for future studies.

The choice of isolation buffer, especially pH, is also a critical factor in the extraction of GDVLNs. According to the results of two independent studies, acidic conditions are considered a favorable environment for the presence and isolation of exosomes and contribute to the enrichment of certain bioactive components in exosomes [21,22]. However, there is a limit to the range of such pH adjustments, depending on the characteristics of the GDVLNs under investigation. When studying potentially acidic compounds in GDVLNs, an alkaline isolation buffer should be used to deprotonate acidic groups [23]. Due to the resulting charge and lipophobicity, these compounds will be unable to cross the membrane barrier and will remain either inside or outside the vesicle, depending on their original location. Conversely, when studying basic compounds, an acidic isolation buffer should be used.

Extraction and purification of GDVLNs is challenging because existing methods often disrupt cells and lead to impurities in the product, such as organelles or membrane structures, such as the plasma membrane [24]. Jang et al. showed that the combined application of ultracentrifugation and the ExoQuick method significantly improved the purity of GDVLNs (up to 83.3%), and that the ExoQuick method resulted in better colloidal stability of the obtained GDVLNs, with an improvement almost double that of the ultracentrifugation method [25]. However, the increase in GDVLN purity was accompanied by a decrease in the total number of GDVLNs, and the combined method reduced the total number of GDVLNs by approximately four-fold compared with the single method. Therefore, this combined method is suitable for the isolation of high-purity and stable GDVLNs from ginseng [25]. As the current extraction and purification techniques for GDVLNs are still challenging, it is difficult to achieve a balance between purity and concentration. Therefore, the main considerations when choosing a method should be the study purpose and problems. Tissue disruption methods are suitable for therapeutic purposes and the large-scale preparation of GDVLNs, whereas obtaining high-purity GDVLNs by combining several extraction methods is more suitable for studying biological mechanisms.

Interestingly, recent studies have reported the detection of choline phosphate in herbal decoctions and its possible entry into human alveolar and gastric cells via the formation of liposomes with small RNA [26]. This suggests that compound herbal tablets may be another source of nanovesicles. Therefore, it is worthwhile to investigate whether Chinese herbal medicinal tablets, such as raw sun-dried ginseng, red ginseng, and black ginseng, which have been treated with different combinations, can also produce GDVLNs with specific properties. One study has shown that black ginseng formed after steam conversion contains fewer polar ginsenosides and is accompanied by an increase in the content of phenolic compounds, reducing sugars, and acidic polysaccharides, showing significantly better results than white and red ginseng [27].

Appropriate storage conditions that ensure the integrity and stability of GDVLNs are essential, as their degradation may result in the loss of their biological activity and ability to enter mammalian cells. Although more precise storage protocols may be required for different types of vesicles, there are currently few studies on the storage of GDVLNs; therefore, some guidance can be obtained from studies of other PDEVs. These studies have shown that some PDEVs can remain in a steady state for >1 month at −80 °C or even 4 °C, but repeated freezing and thawing, including that of GDVLNs, can lead to increased particle size and vesicle fragmentation [28,29]. Based on published data on the resuspension of MDEVs in phosphate-buffered saline (PBS), long-term storage at −80 °C is recommended to prevent the degradation of proteins and nucleic acids. Studies have shown that alginate is very effective in protecting MDEVs at −80 °C. For example, Bosch et al. used alginate-containing PBS as a storage solution for pancreatic β-cell exosomes and significantly reduced the increase in particle concentration and size distribution width during freeze–thaw cycling. No signs of lysis or incomplete vesicles were observed using cryo-electron tomography, and the biological activity of the exosomes was maintained [30]. This suggests that the use of disaccharide stabilizers in storage buffers may be a promising approach for protecting GDVLNs. Additionally, cellular exosomes of mammalian origin can maintain an approximately spherical structure after aerosol drying [31]; however, these data need to be further validated in GDVLNs. In conclusion, there is a need to investigate more efficient extraction methods and storage conditions, and a more detailed evaluation of the function and stability of GDVLNs in different applications is needed to improve the yield and quality of GDVLNs.

## 3. Therapeutic Applications of GDVLNs

Studies on GDVLNs have become possible because of their favorable safety profiles. Many studies have shown that GDVLNs exhibit a broad spectrum of low toxicity. Even after 72 h of treatment at high concentrations (30 μg/mL), GDVLNs showed no cytotoxicity to mouse macrophages. In addition, no significant damage to organs and tissues such as the brain, heart, kidney, liver, lungs, or spleen was observed in mice when GDVLNs were administered orally, intraperitoneally, or intravenously [15]. GDVLNs may not be transferred across the placenta (at least as demonstrated by grapefruit-derived EVs) [32]. Additionally, the safety of GDVLNs in primary human cells was validated. Cho et al. also found that high concentrations of GDVLNs were less toxic than ginseng root extract [16]. The GDVLNs also exhibited good biocompatibility. GDVLNs can be taken up by a wide range of mammalian cell types and are enriched in several organs, depending on the route of administration [15,16,17]. This suggests that a modulatory effect on target cells can be achieved by altering the substances loaded with GDVLNs. In this section, we summarize the role of GDVLNs in the treatment of various human diseases and their potential as endogenous carriers for drug delivery.

### 3.1. Anti-Tumor Effects

Targeting tumor-associated macrophages (TAMs), which are widely distributed in a tumor microenvironment (TME), is a promising strategy for cancer therapy. Based on macrophage plasticity, actively promoting the anti-tumor immune effects of TAM by reprogramming them is another promising therapeutic approach [33]. Cao et al. found that macrophages could take up and phagocytose GDVLNs. In vitro experiments showed that GDVLNs significantly contributed to the conversion of M2-type macrophages to an M1-type phenotype. This conversion was manifested by increased secretion of M1-related cytokines and increased levels of reactive oxygen species (ROS). GDVLNs mediated melanoma cell apoptosis in a murine melanoma model by promoting Th1 immune responses and increasing CD8+ T cell infiltration. Further studies revealed that ceramide in GDVLNs is a key component that exerts immunomodulatory effects depending on Toll-like receptor 4 (TLR4)/MyD88 signaling pathways to orchestrate the shift in the macrophage polarization phenotype [15]. Additionally, Lv et al. found that GDVLNs reduced ARG1 production by reprogramming TAM, ameliorated T cell failure through the mammalian target of rapamycin (mTOR)-T-bet axis, enhanced CD8+ T cell proliferation and activation, and promoted T cell proliferation and activation [34]. Further studies also showed that combining GDVLNs with the monoclonal antibody programmed cell death protein-1 (PD-1) significantly increased the number of infiltrating T cells in colon and breast cancer TAM, showing a shift from a “cold tumor” to a “hot tumor,” and enhanced long-term antigen-specific anti-tumor memory, inducing durable systemic anti-tumor immunity. This reflects the synergistic potential of remodeling the tumor microenvironment. Mechanistically, GDVLNs increase the secretion of the chemokines CCL5 and CXCL9 by reprogramming TAM to attract CD8+ T cells to infiltrate the tumor tissue [18]. Recent studies have confirmed that GDVLNs directly downregulate PD-L1 levels and upregulate major histocompatibility complex-I levels in breast cancer tissues [35]. Thus, the activation of TAMS by GDVLNs may serve as a simple platform to broadly modulate the immunosuppressive TME and optimize immune checkpoint-targeting therapeutic modalities.

GDVLNs directly affect tumor cell proliferation and migration. Yu et al. demonstrated that GDVLNs could induce 4T1 apoptosis in breast cancer cells through the phosphatase and tensin homolog/phosphoinositide 3-kinase/AKT/mTOR and caspase-dependent pathways [35]. In several lung cancer cell lines, GDVLNs inhibit cell migration, invasion, cloning, and adhesion tube formation. This anti-tumor effect may involve the regulation of thymidine phosphorylase expression and pentose phosphate pathway inhibition [18].

The blood–brain barrier (BBB) plays an important role in maintaining homeostasis in the brain. However, in the treatment of intracerebral tumors, the presence of the BBB significantly reduces the efficacy of drugs targeting brain tumors; therefore, it is crucial to develop drugs that can cross the BBB [36]. Kim et al. showed that GDVLNs could cross the BBB after intravenous injection into rats and be taken up by intracerebral tumors. Ptc-mir396f in GDVLNs significantly silenced *c-MYC* expression, thereby inhibiting glioma growth in vivo. GDVLNs downregulated the expression of pro-tumor cytokines in the TME and significantly inhibited the activity of regulatory T cells and tumor-associated fibroblasts [37]. GDVLNs demonstrated an excellent ability to cross the BBB in both in vivo and in vitro experiments, providing a safe and effective platform for the treatment of a wide range of intracerebral disorders (e.g., neurodegenerative diseases, stroke, and intracranial infections) (Figure 2).

### 3.2. Anti-Inflammatory and Antioxidant Effects

Inflammation and oxidative stress function in a mutually reinforcing cascade that drives the development of many diseases and exacerbates pathological processes. GDVLNs, like many ginseng extracts, have been shown to possess potent antioxidant capabilities. Wang et al. found that GDVLNs can diffuse into joints and exhibit anti-inflammatory effects in mice with collagen-induced arthritis (CIA). Mechanistically, a specific pgi-miR6135j in GDVLNs can be delivered to synovial macrophages, thereby reducing the phosphorylation level of the mitogen-activated protein kinase (MAPK) pathway by inhibiting Kirsten rat sarcoma virus expression [38]. Alcoholic liver injury (ALI) is one of the major causes of liver damage and is closely associated with oxidative stress and inflammatory response [39]. Li et al. showed that GDVLNs reduced oxidative stress and inflammatory response by activating the Nrf2/HO1 pathway and blocking the NF-κB pathway, thereby attenuating ALI [40]. Cytokines released during the inflammatory state can promote osteoclast differentiation and activation, exacerbating bone resorption, which is an important cause of osteoporosis development [41]. Seo et al. showed that GDVLNs were enriched with ginsenosides Rb1 and Rg1 and regulated genes for osteoclast maturation by inhibiting the receptor activator of nuclear factor κB ligand-induced IκBα, Jun N-terminal kinase (JNK), and extracellular signal-regulated kinase signaling pathways, showing a more potent effect on osteoclast maturation than the single use of ginsenosides. This study demonstrates the inhibitory effect of GDVLNs on osteoclast differentiation in a mouse model of lipopolysaccharide-induced bone resorption mouse model [42].

GDVLNs exert anti-inflammatory effects on inflammatory bowel disease (IBD). Kim et al. found that GDVLNs had intestinal retention effects for 48 h and effectively suppressed pro-inflammatory cytokine production through inhibition of NF-κB in dextran sodium sulfate-induced colitis. GDVLN treatment also affected the intestinal flora of mice, reducing the proportion of thick-walled bacteria/anthrobacterium phylum and improving the digestive system by balancing the microbiota at the intestinal barrier [43]. Similarly, Song et al. found that GDVLNs act on the TLR4/MAPK and p62/Keap1/Nrf2 pathways to exert anti-inflammatory and antioxidant effects, promote intestinal stem cell proliferation and differentiation, and increase the diversity of intestinal flora, demonstrating a beneficial therapeutic effect in IBD [44]. These studies have shown that GDVLNs can successfully pass through the acidic environment of the stomach and accumulate in the colonic mucosal barrier, so it is worth investigating how these vesicles pass through the “hostile environment” of the gastrointestinal tract. The surface charge of ginger-derived vesicle-like NPs can change from negative to positive in stomach-like solutions, and from positive to negative in small intestine-like solutions [45,46]. It is important to consider this parameter when identifying the function of GDVLNs in vivo or for their use in drug delivery. Micro RNAs and messenger RNAs (mRNAs) naturally present in milk are resistant to acidic conditions and RNase treatment [47]. Therefore, RNAs present in the GDVLNs or enriched during the extraction process may serve as potential components in response to the complex environment of the gastrointestinal tract.

Previous studies have shown that ginseng extracts and ginseng-derived molecules have significant modulatory effects on the skin [37]. Song et al. found that GDVLNs facilitated skin wound healing and attenuated inflammatory responses in a murine skin wound model [48]. Cho et al. isolated GDVLNs from ginseng root and ginseng cell culture supernatants, both of which ameliorated replicative senescence or senescence-associated pigmentation phenotypes in human dermal fibroblasts or ultraviolet B (UVB)-irradiated human melanocytes, respectively, after treatment with GDVLNs, which had anti-aging and anti-pigmentation effects. Similarly, in UVB-irradiated keratinocytes, GDVLNs protected the cells from death and reduced ROS production by downregulating the mRNA expression of pro-apoptotic genes in a dose-dependent manner. Additionally, GDVLNs reduced the mRNA levels of senescence-related genes (matrix metalloproteinases 2 and 3), pro-inflammatory genes (cyclooxygenase-2 and interleukin [IL]-6), and the cellular senescence biomarker p21 by inhibiting activator protein-1 [49].

Promoting neural differentiation is a key step in the transition from “structural repair” to “functional regeneration” of the skin. This is particularly important for treating diabetic feet, traumatic skin defects, and other diseases related to nerve injury [50]. Song et al. found that GDVLNs regulate cell proliferation through the extracellular signal-regulated kinase (ERK) and AKT/mTOR pathways, thereby promoting the migratory capacity of both keratinocytes and vascular endothelial cells [48]. Xu et al. found that genes associated with neural differentiation, maturation, and functionalization are significantly upregulated in bone marrow-derived MSCs (BMSCs) treated with GDVLNs. Xu et al. further constructed photocrosslinked hydrogels loaded with chemokines and GDVLNs, which demonstrated potent efficacy in recruiting and directing the neural differentiation of BMSCs in in vivo experiments in rats and promoted nerve repair and regeneration, as well as efficacy in neovascularization and wound healing [17]. This provides new evidence that plant-derived RNAs can be transferred to mammalian cells and function, suggesting that GDVLNs have great potential in neuroregenerative medicine through their transfer to mammalian stem cells for neural differentiation in vitro and in vivo.

### 3.3. Drug Delivery

Molecular and drug therapies require effective transfection vectors that are nontoxic and do not elicit host immune responses. Therefore, GDVLNs are highly desirable for this purpose. Wang et al. combined GDVLNs with excised autologous tumor-derived membranes to form functional hybrid vesicles. Introduction of functional vesicles improves phagocytosis of autologous tumor antigens by dendritic cells and promotes the maturation of dendritic cells via TLR4. This ultimately activates cytotoxic T lymphocytes to inhibit the recurrence and metastasis of subcutaneous and in situ tumors [51]. Yu et al. constructed GDVLNs hydrogels consisting of gallic acid-treated gelatinized methacryloyl and dimethyloxallyl glycine that exhibited excellent therapeutic effects on wound healing in diabetic rats [52]. Further optimization of the surface modification, drug-carrying capacity, and targeting of GDVLNs, as well as exploring their applications in regenerative medicine and neuroprotection, will surely promote GDVLNs as biomaterials with diverse functions and promising prospects (Figure 3).

## 4. Gintonin: A Potentially Active Substance in GDVLNs

Plants are constantly exposed to biotic stressors during growth. However, they lack an immune system. To combat these stressors, plants have developed specific mechanisms that use secondary metabolites as weapons. These metabolites include phenolic compounds, alkaloids, glycosides, and terpenoids [53,54]. Plants can defend themselves against pathogenic microorganisms by packaging secondary metabolites with antimicrobial activity into their vesicles. Furthermore, the transport of matrix polysaccharides via vesicles is essential for plant cell wall synthesis [55]. Secretion of these vesicles increases in response to bacterial or parasitic infections, indicating that they play a crucial role in the basal immunity and environmental stress response of plants [56,57] (Figure 4). Therefore, GDVLNs may contain polysaccharides, polyphenols, and other unique active substances that potentially have biological effects. Studies have shown that GDVLNs are enriched in various ginsenosides, such as Rg3, Rb1, and Rg1 [15,42]. Additionally, recent studies have revealed that GDVLNs contain 77 compounds, including organic acids and derivatives, amino acids and derivatives, sugars, terpenoids, and flavonoids [40]. Researchers conducted lipidomic analyses and discovered that certain lipid species were present in GDVLNs that were not detected in ginseng root extracts. This indicates that differences in lipid composition between diacylglycerol, some phospholipids, and sphingolipids were enriched by >20-, eight-, and seven-fold, respectively, in GDVLNs compared with the corresponding levels in ginseng root extracts, revealing their unique vesicular properties [19]. Membrane lipids, such as diacylglycerol, can act as secondary messengers by activating protein kinases C and D [58]. Therefore, the differentially expressed lipids in GDVLNs may be important factors influencing the effects of GDVLNs.

Most studies of the efficacy of ginseng have focused on its effects. However, recent evidence suggests that ginseng constituents, including ginsenosides, may not fully demonstrate the systemic effects associated with all potential molecular mechanisms observed in ginseng pharmacology [59]. In addition to ginsenosides, ginseng contains a variety of lysophosphatidic acids (LPAs) that are commonly found in animal and plant systems and have the same chemical structure. LPAs act as ligands for G protein-coupled receptors in animal cells, while in plants, they serve as metabolic intermediates for lipid synthesis or glycerophospholipid storage in plant cell membranes [60,61]. In ginseng, LPAs bind to major latex-like and ribonuclease-like storage proteins to form gintonin [62]. In contrast to ginsenosides, which have no known specific receptors in mammalian cells and require high micromolar concentrations (EC50 or IC50 ≈ 30–97 mM) for delivery [63], gintonin, a first messenger isolated from ginseng, binds cell surface LPA receptors with high affinity and activates cell surface [Ca^2+^]i transients at a range of concentrations less than nanomolar or nanomolar (EC50 ≈ 0. 45–18 nM) to induce changes in [Ca^2+^]i transients in receptor cells. This process is coupled to the enhancement of the phosphatidylinositol-dependent pathway with desensitized P2X1 receptors [64,65], suggesting that gintonin may serve as a potential substance exerting biological effects distinct from those of the ginsenosides.

Gintonin has demonstrated therapeutic efficacy in various disease models (Table 1). Notably, evidence supporting the therapeutic potential of gintonin for treating diseases is similar to that supporting GDVLN’s potential, particularly with regard to its anti-tumor and anti-inflammatory properties. In anti-tumor, the gintonin sensitizes human renal cell carcinoma cells to TRAIL-induced apoptosis by upregulating DR4/5 and apoptotic protein expression [66]. Moreover, gintonin exerts potential anti-tumor effects on melanoma by inducing apoptosis and cell cycle arrest [67]. Furthermore, it has also been demonstrated that gintonin inhibits transforming growth factor beta-induced epithelial-mesenchymal transformation in A549 lung cancer cells [68]. Although no studies have been conducted on the relationship between gintonin and TME, Ray et al. demonstrated that LPAs convert monocytes into macrophages via the AKT/mTOR pathway in both mice and humans [69]. Additionally, they found that LPAs released from rectal cancer cells promote the polarization of TAMS to an M1-like phenotype via LPA receptors [70]. Gintonin blocks the entry of NF-κB into the nucleus by phosphorylating JNK and ERK MAPK in fibroblast-like synoviocytes and suppressing arthritic symptoms in a CIA mouse model [71]. Lee et al. found that oral administration of gintonin reduced serum immunoglobulin E, histamine, IL-4, and interferon gamma levels and significantly reduced eosinophil and mast cell infiltration in the skin and skin inflammation in an atopic dermatitis mouse model [72]. Red ginseng-derived gintonin attenuates UVB-induced cellular senescence through its antioxidant effects by inhibiting the overexpression of cellular β-galactosidase [68]. Importantly, Cao et al. highlighted the importance of the structural integrity of the GDVLN membrane and the proteins it contains for proper functioning [15]. This suggests that the biological effects of GDVLNs may not be limited to a single substance, as described by the researchers in their article, but may involve other compounds as well. Therefore, gintonin may be enriched in GDVLNs and may follow their uptake by recipient cells to exert potential biological effects.

Interestingly, similar to GDVLNs, gintonin affects BBB permeability. For example, Kim et al. demonstrated that gintonin causes morphological changes in primary human brain microvascular endothelial cells (HBMECs) by interacting with LPA1/3 receptors, which increases connectivity gaps and enhances BBB permeability [73]. Jang et al. demonstrated that gintonin reduces cerebral microvascular permeability and preserves microvascular endothelial connectivity proteins in an Alzheimer’s disease mouse model [74]. Effects on the BBB may vary, depending on the route and duration of administration. Choi et al. demonstrated that intraventricular injection of gintonin, but not oral administration, improved the intracerebral delivery of donepezil [75]. Additionally, Kim et al. found time-dependent differences in the expression levels of tight junction proteins in HBMECs [73]. This discrepancy offers an initial reference point for designing dosing regimens that target central nervous system disorders.

## 5. Nanosizing of Ginseng Extracts

### 5.1. Ginsenoside Nanosizing

Ginsenosides are pharmacologically active substances found in ginseng and have been extensively studied. However, their absorption in the gastrointestinal tract is limited by their poor water solubility, gastrointestinal metabolism, and transmembrane transport, resulting in poor bioavailability [76,77]. Additionally, the bioavailability of ginsenosides is affected by the pumping of these compounds out of the cell via cellular efflux transporters [78,79]. The solubility of ginsenosides depends on the number of sugar units in their structure. Ginsenosides contain hydrophobic triterpenes, steroidal glycosides, and hydrophilic sugar side chains in their structure. The proportion of sugar side chains with hydrophilic properties is positively correlated with ginsenoside solubility [80,81]. Ginsenosides that exhibit anticancer activity are mainly secondary (e.g., Rg3, Rh2, and Rg5) [82]. Studies have shown that hydrolysis of ginsenoside sugar chains at different positions (carbon-3 > carbon-6 > carbon-20) results in varying degrees of anticancer activity. Specifically, 20(S)-ginsenosides exhibit greater anticancer activity than their 20(R)-stereoisomers, and ginsenosides with a double bond at C-20(21) demonstrate more potent anticancer activity than those with a double bond at C-20(22) [83]. However, the content of this secondary ginsenoside in ginseng is extremely low, making its direct extraction and application difficult. As a result, it is currently converted from major ginsenosides (such as Rb1, Rb2, and Rf) using β-glucosidase or microbial cell systems [84,85]. These secondary ginsenosides lack sugar chains and are mostly glycoside-dominated, resulting in low water solubility. Improving the delivery of ginsenosides and biomolecular coupling are effective strategies for enhancing their bioavailability.

By combining NPs with ginsenosides, their utilization and side effects can be increased. Additionally, the release rate of ginsenosides from the body can be controlled [86]. Carriers modified with specific fragments have shown improved ability to provide targeted therapeutic results in tumor cells. Lahiani et al. combined ginsenosides Rb1 and Rg1 with carbon nanotubes (CNTs) to enhance their anti-proliferative effects against breast and pancreatic cancers. The application of CNTs resulted in a lower dosage of ginsenosides and enhanced their anti-proliferative effects against tumors [87]. Xu et al. modified glycogen with uronic and α-lipoic acids to obtain the amphiphilic polymer LA-UaGly. This polymer encapsulates ginsenoside Rh2 to form Rh2 NPs. These NPs exhibited strong anti-inflammatory activity, significantly inhibiting the overproduction of nitric oxide and inflammatory cytokines and alleviating the symptoms of ulcerative colitis in model mice [88]. Liposomes are nanoscale agents that can be used as effective clinical cancer therapies. Yu et al. constructed L-Rg3, a liposome loaded with Rg3, which demonstrated stronger tumor cell killing than an Rg3 suspension alone in both in vivo and in vitro experiments [89]. Furthermore, various new methods for delivering ginsenosides using NPs have been developed, including vesicular delivery systems, microemulsions, protein-based nanocarriers, metallic NPs, and bionic NPs [90]. Although NPs offer advantages for drug delivery, it is important to consider the ratio of active ingredients, such as ginsenosides, to nanocarriers. The use of a large number of carriers may lead to toxicity and immunogenicity.

### 5.2. Ginsenoside Excipients

In addition to functioning as loading agents for nanocarriers, the excipient properties of ginsenosides offer novel concepts for the development of nanodelivery systems. Certain secondary ginsenosides can substitute cholesterol to construct nanostructured lipid carriers, stabilize phospholipid bilayers, and form amphiphilic structures due to their ginsenoside-like cholesterol-containing steroidal parent nuclei [91,92]. Ginsenosides are primarily transported to cells via glucose-related transporter proteins (glucose transporter [GLUT]) [93,94]. GLUT is overexpressed in many tumors because of its high dependence of tumor cells on glucose [95]. The liposomes constructed with the participation of ginsenosides can enhance the targeting ability of tumors, according to this feature. Zhu et al. constructed liposomes loaded with paclitaxel (PTX) based on Rg3 (Rg3-PTX-LPs). These liposomes significantly attenuated drug resistance and improved anti-tumor effects compared to conventional cholesterol liposomes. The liposomes recognized GLUT-1 and were specifically distributed to both MCF7/T cells and the TME of breast cancer cells [96]. However, NPs are foreign substances that can be adsorbed by conditioning proteins, such as immunoglobulins and their complex proteins, and can then be phagocytosed and cleared by the reticuloendothelial and mononuclear phagocyte systems, which can affect therapeutic efficacy [97]. Ginsenoside-modified NPs decrease the adsorption of conditioning and complement proteins on a liposome surface, resulting in liposomes exhibiting stealth effects and delayed macrophage uptake [98].

Some ginsenosides are rich in sugar side chains, resulting in improved water solubility [99]. Notably, ginsenoside Rb1 possesses self-assembly properties related to its solubilization. In an aqueous solution, an NP typically forms an aggregate with hydrophilic “head” regions in contact with the surrounding solvent, which isolates the hydrophobic tail regions at the center of the NP. This process increases the water solubility of the hydrophobic drug by localizing it within the hydrophobic region of the NP [100]. Nanosizing of Rb1 for hydrophobic drug delivery is a promising approach. Dai et al. encapsulated hydrophobic natural anticancer drugs, such as betulinic acid, dihydroartemisinin, and hydroxycamptothecin, in Rb1 self-assembled NPs. The drug-loaded NPs exhibited better tumor selectivity and growth inhibition than the free drug, as well as a longer blood half-life, when injected into mice [101]. Zuo et al. utilized the amphiphilic nature of ginsenosides to create carrier-free nanodrugs composed of ginsenosides Rg3 and Rb1 via nanoprecipitation. These nanodrugs demonstrate more potent anti-tumor and anti-invasive effects than free ginsenosides [100].

## 6. Future Perspectives

GDVLNs are considered an attractive option for the future delivery of RNAs, proteins, and therapeutic agents because of their advantages of targeted delivery, safety, and low-cost large-scale production. Despite the continuous progress, further studies are required in several areas related to GDVLNs. Currently, there is an absence of specific protein markers to distinguish the origin of GDVLNs (e.g., whether they are derived from multivesicular bodies or other intracellular vesicles). This hinders the accurate targeting and extraction of functionally specific GDVLNs, limiting the in-depth analysis of their biological functions and targeted applications. Pinedo et al. reported several potential protein markers for characterizing PDEVs [13]. However, further experimental and high-throughput studies are necessary to identify specific protein markers for the classification and characterization of GDVLNs. Furthermore, the existing isolation methods for GDVLNs lack a unified protocol, and the extracted GDVLNs may exhibit variations in structure, composition, and function, which have a deleterious effect on the feasibility of their clinical application. Furthermore, large-scale preparation of GDVLNs necessitates a delicate balance between yield and purity. Existing technologies are still in the nascent stages of cost control, batch stability, and large-scale cultivation, which hinders their industrial application.

To ensure the quality, efficacy, and safety of GDVLNs for medical and pharmaceutical applications, they must be mass-produced using highly reproducible, high-yield, and cost-effective technologies. However, some heterogeneity in the application of GDVLNs may arise because of differences in origin, cultivation methods, and years of ginseng growth. Furthermore, drugs present in the soil or water can be absorbed and accumulated in different parts of the plant. Further studies are necessary to determine whether GDVLNs can naturally encapsulate and transport drugs from the soil or water into human cells. Although these factors may have little effect during the research phase, when GDVLNs are used in small quantities, it is important to focus on these issues when providing large-scale industrial access to GDVLNs to ensure consistency and reliability in medical and pharmaceutical applications.

Despite the existence of patent literature that provides experimental evidence supporting the therapeutic feasibility of GDVLNs for disease treatment, no clinical trials have been initiated, and marketed drugs have yet to be approved. It is important to note that, given the nascent stage of research on GDVLNs, which remains in the preclinical basic exploration phase, the pivotal link in the process of translating laboratory results into clinical applications is still in the preliminary state of advancement. Future studies must concentrate on supplementing systematic evidence at the clinical translational level to comprehensively and scientifically assess the potential for practical applications. In summary, the subsequent evolution of GDVLNs must overcome the impediments posed by foundational mechanisms and technologies. Through the implementation of standardization, engineering, and multidisciplinary research, these networks can transition from laboratory to clinical and industrial applications. This study provides a novel category of natural nanoplatforms for disease treatment and drug delivery.

## Figures and Tables

**Figure 1 antioxidants-14-00961-f001:**
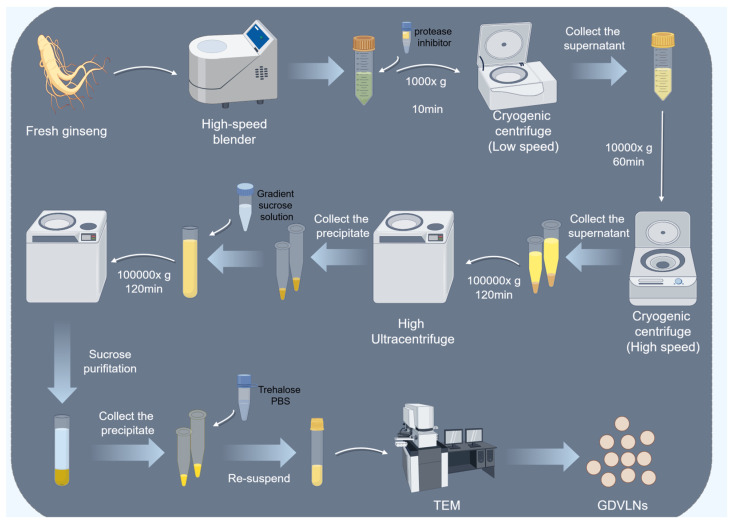
Acquisition of GDVLNs and identification methods. Following the crushing of clean ginseng roots, tissue fragments and cellular residues were removed by one or two low-speed centrifugations. The GDVLNs were then collected by ultra-high-speed centrifugation, followed by purification by density gradient centrifugation. Finally, the morphology was identified by transmission electron microscopy. By Figdraw.

**Figure 2 antioxidants-14-00961-f002:**
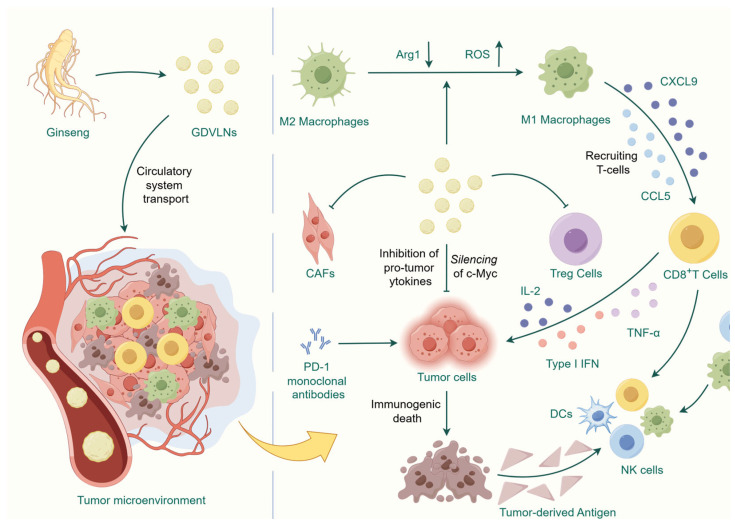
Main mechanism by which GDVLNs exert anti-tumor effects. GDVLNs have been shown to silence genes, such as *C-MYC*, thereby inhibiting tumor cell proliferation. Additionally, GDVLNs have been observed to be taken up by macrophages, which in turn promotes the polarization of M2-type macrophages to M1-type. GDVLNs have been shown to promote the secretion of chemokines, such as CCL5 and CXCL9, by macrophages, which in turn serve to recruit CD8+ T cells into tumor tissues. Additionally, GDVLNs have been observed to down-regulate pro-tumorigenic cytokines within the tumor microenvironment, a process that involves the suppression of Tregs and CAFs. When utilized in conjunction with a PD-1 monoclonal antibody, it has been observed to transform “cold tumors” into “hot tumors,” thereby eliciting anti-tumor effects through the activation of immune responses and direct inhibition of tumor cells. By Figdraw.

**Figure 3 antioxidants-14-00961-f003:**
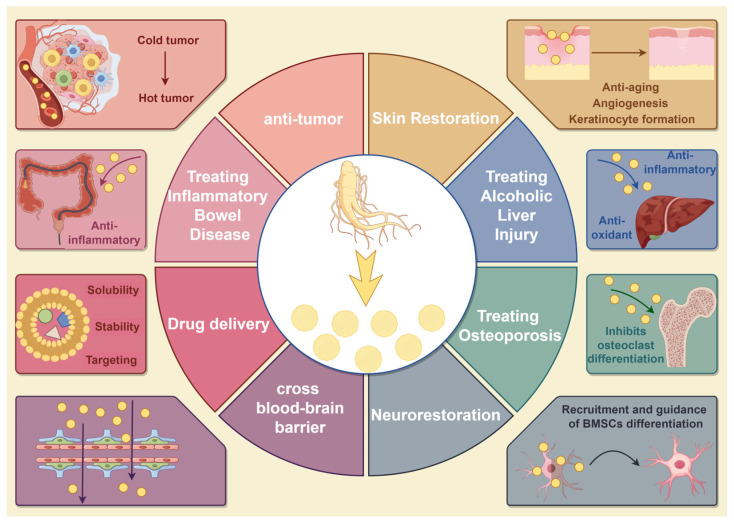
GDVLNs play a role in the treatment of a variety of diseases and have a wide range of applications. GDVLNs possess a variety of biological activities, including anti-tumor (converting “cold tumors” to “hot tumors”), anti-inflammatory, antioxidant, anti-aging, and skin repair properties. Additionally, they promote angiogenesis and have been observed to treat IBD, ALI, and osteoporosis. GDVLNs have also demonstrated the ability to cross the BBB, suggesting potential applications in nerve repair. Furthermore, GDVLNs can function as a drug delivery carrier, highlighting their multifaceted potential in the fields of disease treatment and drug delivery. By Figdraw.

**Figure 4 antioxidants-14-00961-f004:**
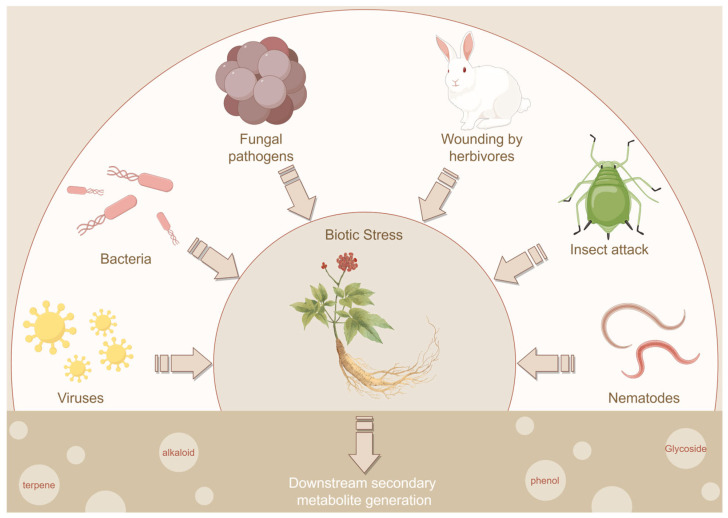
Plants exposed to various biotic stresses, including fungal pathogens, insect attacks, and bacterial or viral infections, have been observed to produce secondary metabolites such as glycosides, alkaloids, phenolics, and terpenoids. These secondary metabolites are synthesized as vesicles and function as defensive molecules, helping plants to combat biotic stresses. By Figdraw.

**Table 1 antioxidants-14-00961-t001:** Potential mechanisms for the therapeutic role of gintonin in multiple disease models.

Diseases/Model	Mechanisms/Function	PMID
Gastric Ulcers	Inhibits the expression of inflammatory cytokines and increases COX-2 and LPA5 receptor expression and PGE2 levels	38069044
Wound-Healing	Epidermal growth factor receptor activation and heparin-binding EGF-like growth factor release	37762395
	LPA receptor activation and / or VEGF release	34576317
Skin ageing	Inhibition of beta-galactosidase overexpression	37299538
Lung cancer	Inhibition of cancer cell metastasis by maintaining the integrity of cell-cell junctions	37653930
Epilepsy	Decreased levels of glial cell activation and expression of pro-inflammatory cytokines/enzymes and increased levels of Nrf2 antioxidant response	37252272
Neuroprotection	Activation of Akt and CREB stimulates dendritic growth of striatal neurons.	36385759
	Activation of the LPA1 receptor-BDNF-TrkB-Akt signaling pathway reduces oxidative stress in neuronal cells	34299412
Sarcopenic obesity	Promoting energy expenditure and reducing skeletal muscle atrophy	35600770
Arthritis	Inhibiting the activation of NF-κB and reducing the expression of inflammatory factors by mediating the phosphorylation of JNK, ERK, and MAPK	34803427
Cancer cachexia	Reduce TNF-α-induced oxidative stress and muscle atrophy by reducing ROS and inhibiting inflammation-related genes.	34798386
Sarcopenia	Restoring age-related immune homeostasis by maintaining the T cell compartment and regulating inflammatory biological responses	34764729
Amyotrophic lateral sclerosis	The level of oxidative stress and the activation of immunoreactive glial cells were reduced, and the expression level of LPA1 receptor was restored	34025132
Alzheimer’s disease	Attenuated cerebral microvascular permeability and disruption of microvascular endothelial junction proteins	33717253
Obesity	Decreased expression of pro-adipogenic and adipogenic factors to reduce lipid accumulation and increase lipolysis and thermogenesis	32679738
Lead poisoning	Inhibition of neuronal apoptosis and attenuation of oxidative stress and inflammation in the brain	32131481
Heat stress	Reducing oxidative stress and inflammatory damage	32106493
Hair growth	Stimulated release of vascular endothelial growth factor	32095099
Mercury poisoning	Reduced methylmercury-induced neurotoxicity and oxidative stress and increased its removal	32013120

## Data Availability

Not applicable.

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
