# Peer review of "Ginseng Nanosizing: The Second Spring of Ginseng Therapeutic Applications"

_antioxidants, 2025, doi:10.3390/antiox14080961_

Round 1
Reviewer 1 Report
This review focuses on ginseng-derived vesicle-like nanoparticles (GDVLNs) and covers key areas such as extraction and purification methods, therapeutic potential, active ingredients, and nanosized applications of ginseng extracts. It systematically reviews the research progress of GDVLNs in antitumor, anti-inflammatory, and antioxidant properties, as well as drug delivery. It elaborates on the mechanisms of action of key active ingredients, such as ginsenosides and gintonin, and nanosizing strategies, providing a more comprehensive reference for research in this field. However, some improvements could be made, such as making the transitions between chapters more natural and condensing some of the mechanism descriptions to enhance readability. Overall, however, the review effectively summarizes the current status and prospects of ginseng nanoparticles, and it is recommended for revision and acceptance.
- In Chapter 2, "Acquisition of GDVLNs," a brief summary of the structural differences (e.g., diameter, morphology) of GDVLNs caused by different extraction methods could be added at the end to improve the completeness of the content.
- In Section 3.1, "Anti-tumor Effects," when describing the combination of GDVLNs with PD-1 antibodies, a summary of the potential for synergistic therapy could be added after "Enhancement of Long-Term Antigen-Specific Anti-tumor Memory."
- Chapter 4, which discusses the differences in the actions of Gintonin and GDVLNs at the blood-brain barrier, could be supplemented with a summary after the description of the route of administration.
- In sections 5.1 and 5.2, transition sentences could be added where the two sections meet to make them more logically coherent when discussing the nanosizing of ginsenosides.
- The keyword "nanosizing" could be added.
- Unify the subheading "Anti-inflammatory and Antioxidant" in Section 3.2 as "Anti-inflammatory and Antioxidant Effects," consistent with the subheading in Section 3.1.
- The chapter numbering is duplicated, so "5. Future Perspectives" should be adjusted to "6. Future Perspectives."
Reviewer 2 Report
I appreciate the opportunity to review this manuscript. While the topic is timely and relevant, I strongly recommend the following revisions to enhance its scientific rigor and overall quality:
-
Literature Search Strategy: Although this is a review article, it is essential to outline a clear and systematic strategy for identifying and selecting the included studies. Please specify the databases consulted, search terms used, and inclusion/exclusion criteria. This will strengthen the transparency and reproducibility of your review.
-
Future Perspectives: Include a well-developed final section discussing key open questions, current technological challenges, and emerging research directions. This should be explicitly linked to potential biomedical or industrial applications of the reviewed technologies.
-
Translational Relevance: Please indicate whether there are any ongoing clinical trials or registered patents related to the discussed technologies. This information is important to fully assess the current state of the art and potential for practical implementation.
-
Figures: Improve the clarity and descriptive quality of all figures. Additionally, indicate the source of each figure (i.e., whether they are original or adapted from previous work) to avoid any conflict of interest or copyright issues.
No more comments.
Reviewer 3 Report
The paper reviews the extraction and purification process of ginseng-derived vesicle-like nanoparticles (GDVLNs), their therapeutic potentials, and the active ingredients in GDVLNs that may exert pharmacological activities. This is an interesting topic, but the manuscript has some weaknesses that should be eliminated.
While the manuscript is labeled and presented as a review, it lacks a critical component expected in both narrative and systematic reviews: a description of the methodology used to collect and select the cited literature.
Specifically, the authors have not indicated:
- Which databases were searched (e.g., PubMed, Web of Science, Scopus),
- What keywords or Boolean operators were used,
- Whether there were any inclusion or exclusion criteria,
- The time frame of literature considered,
- Or any process by which the literature was screened for relevance.
This omission significantly limits the transparency and reproducibility of the review. Even in the case of a narrative review, it is standard practice to include at least a brief methodology statement explaining how the literature was gathered. Without such information, it is difficult to assess the comprehensiveness, objectivity, or potential bias in the selection of studies included in the manuscript.
I recommend the authors include a dedicated methodology section or at least a paragraph within the introduction detailing the scope of the literature search, databases used, and general inclusion criteria. This will improve the scholarly rigor of the manuscript and more accurately justify its classification as a review.
Some points are reiterated in slightly different terms across sections. For example, the therapeutic potential and delivery mechanisms of GDVLNs are revisited in both the “Applications” and “Future Perspectives” sections. Please focus on the essentials and avoid repetitions.
Several references are used repeatedly and support general claims (e.g., studies on GDVLNs and their safety or drug delivery potential). Some citations are general or loosely linked to claims - although in this case, many references appear legitimate and specific.
Please use first the general reference then the details (Two examples are given under detailed comments).
Overall, the work provides a nice overview, but please fulfill the requirements for a review.
“the volume of conditioned medium required for the 4.03 × 1012 EVs obtained from MSCs by Verena Börger et al. was approximately 4300 mL, whereas the number of EVs isolated from carrots was approximately 3.24 × 1011/g”. Please give comparable measures and not number/mL compared with number/g.
Line 74: In the current study, ginseng root was typically crushed in a blender to obtain plant sap. Is [14] a typical study or is it better first to introduce different methods before you describe details?
Line 102: The choice of isolation buffer, especially pH, is a critical factor in the extraction of GDVLNs, [21, 22, 23] [23] is a review! Again, first the general reference [23] then the details. Please consider this also in the other paragraphs.
Round 2
Reviewer 3 Report
Thank you for considering my comments. In my opinion, the review provides a comprehensive overview of Ginseng NPs and their application.
no additional comments